# MV2MAE: Self-Supervised Video Pre-Training with Motion-Aware Multi-View Masked Autoencoders

**Ketul Shah**[†]                                                        *ketuls27@gmail.com*
*Johns Hopkins University*

**Robert Crandall**                                                    *rcrandal@amazon.com*
*Amazon*

**Jie Xu**[*]                                                                *jiexuwj@gmail.com*
*Sony AI*

**Peng Zhou**                                                    *pengzhou@terpmail.umd.edu*
*Amazon*

**Vipin Pillai**                                                        *pilvipin@amazon.com*
*Amazon*

**Marian George**[*]                                                    *mariang@google.com*
*Google*

**Mayank Bansal**                                                    *maybans@amazon.com*
*Amazon*

**Rama Chellappa**                                                        *rchella4@jhu.edu*
*Johns Hopkins University*

**Reviewed on OpenReview:** *https://openreview.net/forum?id=nqt35xJywK*

## Abstract

Videos captured from multiple viewpoints can help in perceiving the 3D structure of the world and benefit computer vision tasks such as action recognition, tracking, etc. In this paper, we present MV2MAE, a method for self-supervised learning from synchronized multi-view videos, built on the masked autoencoder framework. We introduce two key enhancements to better exploit multi-view video data. First, we design a cross-view reconstruction task that leverages a cross-attention-based decoder to reconstruct a target viewpoint video from source view. This helps in effectively injecting geometric information and yielding representations robust to viewpoint changes. Second, we introduce a controllable motion-weighted reconstruction loss which emphasizes dynamic regions and mitigates trivial reconstruction of static backgrounds. This improves temporal modeling and encourages learning more meaningful representations across views. MV2MAE achieves state-of-the-art results on the NTU-60, NTU-120 and ETRI datasets among self-supervised approaches. In the more practical transfer learning setting, it delivers consistent gains of +2.0 – 8.5% on NUCLA, PKU-MMD-II and ROCOG-v2 datasets, demonstrating the robustness and generalizability of our approach. Code: https://github.com/kshah33/mv2mae

---

[†]Work completed during internship at Amazon.
[*]Work completed while at Amazon.

# 1 Introduction

Multiple viewpoints of the same event are crucial to its understanding. Humans move around and obtain different viewpoints of objects and scenes, and develop a representation robust to viewpoint changes (Isik et al., 2018). Different viewpoints often have very different appearance, which can help address challenges due to occlusion, lighting variations and limited field-of-view. In many real world scenarios, we have videos captured from multiple viewpoints, *e.g.* sports videos (Saito et al., 2004), elderly care (Jang et al., 2020), self-driving (Yogamani et al., 2019), complex robotic manipulation tasks (Seo et al., 2023a) and security videos (Corona et al., 2021). Learning a robust pre-trained model from large amounts of unlabeled synchronised multi-view data is of significant value for these applications. Such a model which is aware of the 3D geometry will be robust to changes in viewpoint and can be effectively used as a foundation for downstream finetuning on smaller datasets for different tasks.

There has been significant progress in video self-supervised learning (Schiappa et al., 2023) (SSL) for the single-view case, *i.e.* where synchronized multi-view data is not available. Recently, Masked Autoencoders (MAEs) as a paradigm for self-supervised learning has seen growing interest, and it has been successfully extended to video domain (Feichtenhofer et al., 2022; Tong et al., 2022; Wang et al., 2023a). MAE-based methods achieve superior performance (Tong et al., 2022) on standard datasets such as Kinetics-400 (Kay et al., 2017) and Something-Something-v2 (Goyal et al., 2017), compared to contrastive learning methods (Feichtenhofer et al., 2021). However, existing MAE-based pre-training approaches are not explicitly designed to be robust to viewpoint changes. View-invariant learning from multi-view videos has been widely studied using NTU (Shahroudy et al., 2016; Liu et al., 2020a) and ETRI (Jang et al., 2020) datasets. However, most of these methods use 3D human pose, which is difficult to accurately capture for in-the-wild scenarios. There has been a growing interest in RGB-based self-supervised learning approaches leveraging multi-view videos (Parameswaran & Chellappa, 2006; Das & Ryoo, 2023; Vyas et al., 2020; Li et al., 2018), facilitated by the availability of large-scale multi-view datasets (Shahroudy et al., 2016; Liu et al., 2020a; Jang et al., 2020). ViewCLR (Das & Ryoo, 2023), which achieves state-of-the-art results among SSL methods, introduces a latent viewpoint generator as a learnable augmentation for generating positives in a contrastive learning (Chen et al., 2020) framework. However, this method is memory intensive as it requires storing two copies of the feature extractor and two queues of features, while also requiring multi-stage training. In contrast, the recent success of MAEs for video SSL motivates us to explore its potential in the *multi-view* video SSL scenario.

In this paper, we aim to learn self-supervised video representations that are robust to viewpoint shifts. Humans learn a representation robust to viewpoint variations for tasks such as action recognition and are able to *visualize how an action looks from different viewpoints* (Isik et al., 2018). Motivated by this, we design the task of using one viewpoint to predict the appearance from a different viewpoint, and integrate it in the MAE framework. More specifically, given a video of an activity from one viewpoint, it is converted to patches and a high fraction of the patches are masked out. The visible patches are encoded, which the decoder uses (along with MASK tokens for missing patches) to reconstruct the given video. We introduce an additional cross-view decoder, which is tasked with reconstructing the masked patches of a target viewpoint by using the visible regions from source view. This requires the model to understand the geometric relations between different views, making the pre-trained model robust to viewpoint shifts. Another challenge with MAE in videos is temporal redundancy, which makes it easier to reconstruct the static, background regions by simply copy pasting from adjacent frames where those are visible. Existing solutions for this problem involve specialized masking strategies using extra learnable modules (Bandara et al., 2023; Huang et al., 2023) or tube masking (Tong et al., 2022; Wang et al., 2023a), which are not effective in certain scenarios, *e.g.* when motion is localized in a small region of the frame. We propose a simple solution, without introducing additional learnable parameters, by modifying the reconstruction loss to focus on moving regions. We can control the relative weights of moving and static regions using a temperature parameter.

We perform pre-training experiments on three multi-view video datasets: NTU-60 (Shahroudy et al., 2016), NTU-120 (Liu et al., 2020a), ETRI (Jang et al., 2020). Our method achieves SOTA accuracy on these action recognition benchmarks in the full fine-tuning protocol. More notably, the robustness of our representation is shown in the transfer learning results on smaller datasets. We achieve SOTA results on NUCLA (Wang

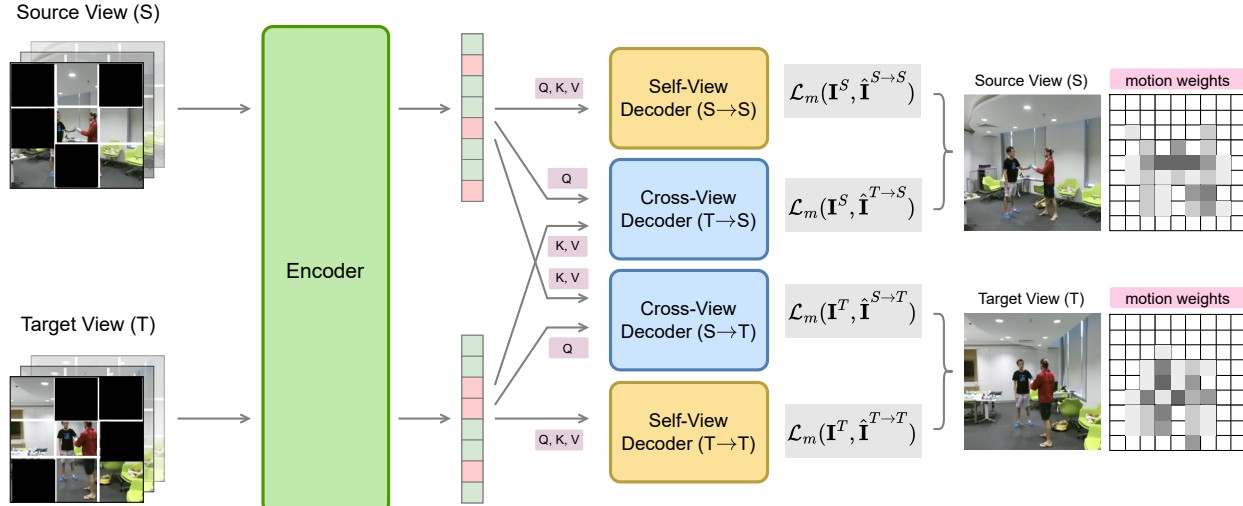

Figure 1: **Multi-View Video Masked Autoencoder (MV2MAE).** Given synchronized videos from a source (S) and target (T) viewpoint, we first tokenize and encode visible patches using a shared encoder. We introduce a cross-view decoder to reconstruct one viewpoint from the other, while the self-view decoder reconstructs each view independently. The proposed motion-weighted reconstruction loss $\mathcal{L}_m(\mathbf{I}, \hat{\mathbf{I}})$, computed using the input clip $\mathbf{I}$ and reconstructed clip $\hat{\mathbf{I}}$, emphasizes dynamic regions and reduces the effect of trivial reconstruction of static background regions. The motion weights are derived from the input clip. Decoders in the same color share weights.

et al., 2014), ROCOG-v2 (Reddy et al., 2023) and PKU-MMD-II (Liu et al., 2017) datasets in the transfer learning setting.

Our main contributions can be summarized as follows:

- We present MV2MAE, a self-supervised pre-training approach explicitly designed for synchronized multi-view videos, achieving state-of-the-art results among SSL methods on NTU-60, NTU-120, and ETRI benchmarks, as well as strong transfer gains (+2.0–8.5%) on NUCLA, PKU-MMD-II, and ROCOG-v2. We also show synthetic multi-view data as a scalable alternative for pre-training data.

- We introduce a dedicated cross-view decoder that reconstructs target viewpoint video from source view, forcing the encoder to capture 3D geometry and learn viewpoint-robust representations.

- We design a simple yet effective motion-aware loss that emphasizes moving regions while downweighting static backgrounds, mitigating trivial reconstruction and substantially improving temporal representation learning.

## 2 Related Work

### 2.1 Self-Supervised Learning from Videos

**Pretext Learning.** Many pretext tasks have been proposed for learning self-supervised video representations, initially inspired from the progress in SSL for images. Tasks such as video rotation prediction (Jing et al., 2018), solving spatio-temporal jigsaw (Ahsan et al., 2019), predicting motion and appearance statistics (Wang et al., 2019) were direct extensions of their image counterparts, and showed impressive performance. Methods leveraging the temporal order in videos for constructing pretext tasks such as frame ordering (Xu et al., 2019) and odd-one-out learning (Fernando et al., 2017) were also proposed. These methods were outperformed by contrastive learning approaches.

**Contrastive Learning.** These methods create augmented versions of the input (positives) which preserve the semantic content of the input. The contrastive loss is used to pull these closer together in the feature space, while simultaneously pushing them away from other samples (negatives). Numerous ways of generating positive pairs were proposed such as using random clips from the same video, clips of different frame rates (Wang et al., 2020), choosing nearby clips (Qian et al., 2021), and using optical flow (Han et al., 2020), among others.

**Masked Video Modeling.** Recently, masked video modeling has emerged as a promising area for SSL. Methods such as BEVT (Wang et al., 2022), MaskFeat (Wei et al., 2022), VideoMAE (Tong et al., 2022), MAE-ST (Feichtenhofer et al., 2022) show superior performance on the standard video self-supervised learning benchmarks. Different reconstruction targets have been studied, such as MVD (Wang et al., 2023b) which uses distillation from pre-trained features, and MME (Sun et al., 2023) which reconstructs motion trajectories. To tackle trivial reconstruction solution via copy-paste in videos, which becomes an issue due to high redundancy, different masking strategies have been proposed. MGMAE (Huang et al., 2023) uses motion-guided masking based on motion vectors, VideoMAE (Tong et al., 2022) proposed using tube masking, AdaMAE (Bandara et al., 2023) introduces a neural network for mask sampling. Orthogonal to these, we propose to tackle the issue by using a motion-weighted reconstruction loss. Moreover, unlike the proposed approach, existing MAE pre-training approaches are not explicitly designed to be robust to viewpoint shifts. Beyond action recognition, the masked modeling paradigm has also been explored in adjacent fields such as robotics, where multi-view masked world models (Seo et al., 2023b) have been utilized for visual robotic manipulation.

### 2.2 Multi-View Action Recognition

Early works in this area designed hand-crafted features which are robust to viewpoint shifts (Parameswaran & Chellappa, 2006; Rao et al., 2002; Xia et al., 2012). Many unsupervised learning approaches have been proposed for learning representations robust to changes in viewpoint. A large number of methods leverage 2D/3D human pose information (Shah et al., 2022; Zhou et al., 2025), which greatly aids in achieving robustness to viewpoint variations. Methods based on RGB modality (Das & Ryoo, 2023; Li et al., 2018; Vyas et al., 2020) have gained increasing popularity. These can be broadly divided into two categories:

One trend is to enforce the latent representations of different viewpoints to be close. Along this line, (Zheng et al., 2012) follows a dictionary learning approach and encourages videos of different views to have the same sparse representation. (Rahmani & Mian, 2015) fits a 3D human model to a mocap sequence and generates videos from multiple viewpoints, which are forced to predict the same label. More recently, methods based on contrastive learning have been proposed such as ViewCLR (Das & Ryoo, 2023) which achieves remarkable performance. They add a latent viewpoint generator module which is used to generate positives in the latent space corresponding to different views.

Another line of work uses one viewpoint to predict another. (Li et al., 2018) uses cross-view prediction in 3D flow space by using depth as an additional input to provide view information. Their approach also uses a gradient reversal layer for achieving robustness to view changes. (Vyas et al., 2020) uses the encoded source view features to render same video from unseen viewpoint and a random start time. Their approach hence needs to be able to predict across time and viewpoint shifts. They leverage a view embedding which requires information of camera height, distance and angle. In contrast to these approaches which rely on view embedding or depth for providing viewpoint information, the view information is inherently available in the visible patches of the source and target viewpoints in our approach. While CrossMAE (Guo et al., 2024) also uses cross-attention in the decoder, it does not explicitly address synchronized multi-view geometry, and RGB-based viewpoint-agnostic methods such as 3D TRL (Shang et al., 2022) rely on temporal relational learning rather than cross-view reconstruction, which is central to our approach.

## 3 Method

### 3.1 Preliminary: Masked Video Modeling

Here we revisit the MAE framework for videos. Given a video, we first sample $T$ frames with stride $\tau$ to get the input clip: $\mathbf{I} \in \mathbb{R}^{C \times T \times H \times W}$. Here, $H \times W$ is the spatial resolution, $T$ denotes the number of frames sampled, and $C$ is the number of input (RGB) channels. The standard MAE architecture has three main components: tokenizer, encoder & decoder.

**Tokenizer.** The input clip is first converted into $N$ patches using a patch size of $t \times h \times w$, where $N = \frac{T}{t} \times \frac{H}{h} \times \frac{W}{w}$. The tokenizer returns $N$ tokens of dimension $d$ by first linearly embedding these $N$ patches. This is implemented in practice using a strided 3D convolution layer. Next, we provide position information to these tokens by adding positional embeddings (Vaswani et al., 2017).

**Encoder.** A high fraction of these $N$ tokens are dropped with a masking ratio $\rho \in (0, 1)$. Different masking strategies (Tong et al., 2022; Bandara et al., 2023; Huang et al., 2023) have been explored for choosing which tokens to mask out. Next, the remaining small fraction of visible tokens are passed through the encoder ($\Phi_{\texttt{enc}}$) to obtain latent representations. The encoder is a vanilla ViT (Dosovitskiy et al., 2020) with joint space-time attention (Tong et al., 2022). These latent representations need to capture the semantics in order to reconstruct the masked patches.

**Decoder.** The encoded latent representations of the visible patches are concatenated with learnable [MASK] tokens corresponding to masked-out patches, resulting in combined tokens $\mathbf{Z}$. The positional embeddings are then added for all tokens, and passed through a light-weight decoder ($\Phi_{\texttt{dec}}$) to get the predicted pixel values $\hat{\mathbf{I}} = \Phi_{\texttt{dec}}(\mathbf{Z})$.

The loss function is the mean squared error (MSE) between the reconstructed values and the normalized pixel values (Feichtenhofer et al., 2022; Tong et al., 2022), for masked patches $\Omega$.

$$\mathcal{L}(\mathbf{I}, \hat{\mathbf{I}}) = \frac{1}{\rho N} \sum_{i \in \Omega} |\mathbf{I}_i - \hat{\mathbf{I}}_i|^2 \tag{1}$$

### 3.2 Cross-View Reconstruction

The goal of cross-view reconstruction is to predict the missing appearance of a video in target viewpoint using the visible patches of video from source viewpoint. Being able to extrapolate across viewpoints requires understanding the geometric relations between different viewpoints, making it an effective task for learning representations robust to viewpoint variations.

As shown in Figure 1, consider two synchronized videos of an activity, $\mathbf{I}^S$ and $\mathbf{I}^T$, from source view ($S$) and target view ($T$) respectively. We first tokenize, mask and encode the visible tokens for each video separately using a shared encoder $\Phi_{\texttt{enc}}$. We then introduce a cross-view decoder ($\Phi_{\texttt{dec}}^{\texttt{cross-view}}$) which additionally uses the visible tokens in the source view to reconstruct the target viewpoint video, $\hat{\mathbf{I}}^{S \to T} = \Phi_{\texttt{dec}}^{\texttt{cross-view}}(\mathbf{Z}^T, \mathbf{Z}^S_{\texttt{vis}})$. More specifically, each block of the cross-view decoder consists of cross-attention and self-attention layers, followed by a feed-forward layer. The tokens from the target view attend to the visible source view tokens using cross-attention, and then to each other using self-attention. Moreover, the standard decoder ($\Phi_{\texttt{dec}}$) is used to reconstruct video from each viewpoint independently $\hat{\mathbf{I}}^{v \to v} = \Phi_{\texttt{dec}}(\mathbf{Z}^v)$ for $v \in \{S, T\}$. Figure 5 visualizes the cross-view reconstruction quality and cross-attention maps, which demonstrates that the model learns to focus on relevant regions across viewpoints.

A key aspect of methods using the cross-view prediction paradigm is how the viewpoint information is provided: (Vyas et al., 2020) conditions the decoder on a viewpoint embedding, while some approaches (Li et al., 2018) use extra modalities such as depth to provide information about the target viewpoint. In contrast to these, in our approach, the visible patches provide the required target viewpoint information. The *amount* of view information we want to provide can be easily varied by changing the masking ratio.

### 3.3 Motion-Weighted Reconstruction Loss

A given video can be decomposed into static and dynamic regions. Static regions typically involve scene background and objects which do not move throughout the video. Patches from such regions are trivial to reconstruct (Sun et al., 2023; Bandara et al., 2023) due to temporal redundancy in videos. In order to deal with this, we offer a simple solution by re-weighting the reconstruction loss of each patch proportional to the amount of motion within that patch. The motion weights used for re-weighting are obtained using frame difference for simplicity. Note that other motion features such as optical flow, motion history image, etc can be used in place of frame difference, but frame difference is extremely fast to compute. In order to get the final weights, we take the norm of frame difference within each patch, and apply temperature softmax over all tokens. We can control the extent to which to focus on the moving regions by controlling the temperature parameter. The higher the temperature value, the more uniform the resulting weights. Examples of motion weights overlaid on the original frames for different temper-

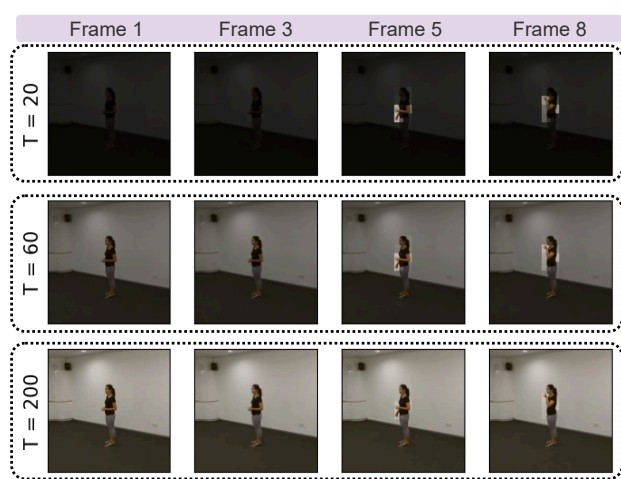

Figure 2: **Motion weights with varying temperature.** Each row shows motion weights overlaid on the input frames for a different temperature. Higher temperature increases the weight on static and background regions.

ature values are shown in Figure 2. PyTorch-style code for computing the motion weights for a clip is provided in the supplementary. The final motion-weighted reconstruction loss ($\mathcal{L}_\mathrm{m}$) is given below, where $w_i(\tau)$ is the weight for $i^\mathrm{th}$ patch with temperature $\tau$:

$$\mathcal{L}_\mathrm{m}(\mathbf{I}, \hat{\mathbf{I}}; \tau) = \frac{1}{\rho N} \sum_{i \in \Omega} w_i(\tau) \times |\mathbf{I}_i - \hat{\mathbf{I}}_i|^2 \tag{2}$$

### 3.4 Pre-training and Evaluation

**Pre-training.** To sum up, we first sample two videos from a set of synchronized videos of an activity, $\mathbf{I}^S$ from source view and $\mathbf{I}^T$ from target view. The encoder, self-view decoder and cross-view decoder are pre-trained using the overall loss given below:

$$\mathcal{L} = \underbrace{\mathcal{L}_\mathrm{m}(\mathbf{I}^S, \hat{\mathbf{I}}^{S \to S}) + \mathcal{L}_\mathrm{m}(\mathbf{I}^T, \hat{\mathbf{I}}^{T \to T})}_{\text{self-view reconstruction}} + \underbrace{\lambda \mathcal{L}_\mathrm{m}(\mathbf{I}^T, \hat{\mathbf{I}}^{S \to T}) + \lambda \mathcal{L}_\mathrm{m}(\mathbf{I}^S, \hat{\mathbf{I}}^{T \to S})}_{\text{cross-view reconstruction}} \tag{3}$$

Here, $\hat{\mathbf{I}}^{S \to T}$ denotes reconstructed video from target view using source view, and so on. $\lambda$ is the weight for cross-view reconstruction loss and is set to 1.

**Evaluation.** Following prior work (Li et al., 2018; Kim et al., 2022; Vyas et al., 2020; Das & Ryoo, 2023), we evaluate our pre-trained models on the task of action recognition using two settings: 1) full fine-tuning on the same datasets and 2) transfer learning on smaller datasets. We discard the decoders, and attach a classifier head which uses the global average pooled features for classification. For testing, we sample 5 temporal clips, and use 10 crops from each following (Das & Ryoo, 2023), taking their average for the final prediction.

## 4 Experiments

We evaluate our approach on several common multi-view video datasets: NTU60 (Shahroudy et al., 2016), NTU120 (Liu et al., 2020a), ETRI (Jang et al., 2020), NUCLA (Wang et al., 2014), PKU-MMD (Liu et al., 2017), and ROCOG (Reddy et al., 2023). For NTU and ETRI, we achieve state-of-the-art results among SSL methods by pre-training and fine-tuning on the target domain. On NUCLA, PKU-MMD, and ROCOG, we demonstrate excellent transfer learning performance by pre-training only on NTU, and fine-tuning on the target dataset.

### 4.1 Datasets

**NTU RGB+D 60.** (Shahroudy et al., 2016) is a large-scale multi-view action recognition dataset, consisting of 56,880 videos from 60 distinct action classes. These videos were recorded from 40 subjects using Kinect-v2. Each activity instance is simultaneously captured from three viewpoints. The dataset consists of two benchmarks outlined in (Shahroudy et al., 2016): (1) Cross-Subject (xsub) and (2) Cross-View (xview). In the cross-subject benchmark, the 40 subjects are divided into training and testing sets, with 20 subjects in each. In the cross-view scenario, videos from cameras 2 and 3 are used for training, while testing is performed on videos from camera 1. This corresponds to using the front view and the $\pm 90°$ views for training, whereas using the intermediate $\pm 45°$ views for testing.

**NTU RGB+D 120.** (Liu et al., 2020a) is an extended version of the NTU-60 dataset containing 114,480 videos spanning 120 action categories. Our evaluation follows the established protocols outlined in (Liu et al., 2020a): (1) Cross-Subject (xsub) and (2) Cross-Setup (xset). In the cross-subject scenario, subjects are partitioned into training and testing groups, while in the cross-setup setting, the data is divided into training and testing subsets based on the setup ID.

**ETRI.** (Jang et al., 2020) is another large-scale multi-view action recognition dataset consisting of activities of daily living for elderly care. It has 112,620 videos captured from 55 action classes. All activity instances are recorded from 8 synchronized viewpoints. (Jang et al., 2020) describes a cross-subject benchmark which we use to evaluate our approach.

### 4.2 Implementation Details

We sample a clip of 16 RGB frames with a stride of 4 from each video. We downsample the resolution of frames to 128×128 following (Das & Ryoo, 2023). During pre-training, we only apply random resized crops as augmentation. We use a temporal patch size of 2 and a spatial patch size of 16×16, which results in 512 tokens. A masking ratio of 0.7 is used unless otherwise specified. We choose fixed sinusoidal spatio-temporal positional embedding following (Tong et al., 2022; Bandara et al., 2023). All of our experiments use the vanilla ViT-S/16 (Touvron et al., 2021) architecture as the encoder, trained using AdamW optimizer (Loshchilov & Hutter, 2017). The pre-training is carried out for 1600 epochs. Please refer to the supplementary material for additional details.

### 4.3 Comparison with state-of-the-art

We compare our approach with prior supervised and self-supervised approaches on the cross-subject (xsub) and cross-view (xview) benchmarks of the commonly used NTU-60 and NTU-120 datasets. We also present our results on the ETRI dataset, which only has a cross-subject benchmark.

Table 1 and Table 2 show results on the NTU-60 and NTU-120 datasets. We outperform all previous *self-supervised methods* based on RGB, Flow or Pose modality on both cross-view and cross-subject benchmarks of the two datasets. In the xsub setting, we see an improvement of +0.3% and +1.2% on NTU-60 and NTU-120 respectively, and in the xview setting, we observe an improvement of +1.8% and +0.9% respectively. ViewCLR (Das & Ryoo, 2023) uses a MoCo (Chen et al., 2020) framework which requires large batch-sizes for convergence (Chen et al., 2021a; He et al., 2022). Vyas *et al.* (Vyas et al., 2020) uses a cross-view prediction paradigm but underperforms (86.3% vs 95.9% on xview and 82.3% vs 90.0% on NTU60 xsub) despite using more parameters ($\sim$72M vs $\sim$22M). Unlike their approach which relies only on learnt viewpoint embeddings

| Method | Modality | Resolution | # Frames | NTU-60 (%) | |
| --- | --- | --- | --- | --- | --- |
| | | | | xview | xsub |
| *Supervised Methods* | | | | | |
| STA-Hands (Baradel et al., 2017) | RGB+Pose | 299×299 | 20 | 88.6 | 82.5 |
| Separable STA (Das et al., 2019) | RGB+Pose | – | 64 | 94.6 | 92.2 |
| ESE-FN (Shu et al., 2022) | RGB+Pose | – | 64 | 96.7 | 92.4 |
| VPN (Das et al., 2020) | RGB+Pose | – | 64 | 96.2 | 93.5 |
| VPN++ (Das et al., 2021) | RGB+Pose | – | 64 | 99.1 | 96.6 |
| 3DA (Kim et al., 2023) | RGB+Pose | 224×224 | 12 | 97.9 | 94.3 |
| PoseC3D (Duan et al., 2022) | RGB+Pose | – | 32 | 99.6 | 97.0 |
| DA-Net (Wang et al., 2018) | RGB | – | – | 75.3 | – |
| Zhang *et al.* (Zhang et al., 2018) | RGB | – | – | 70.6 | 63.3 |
| Glimpse Clouds (Baradel et al., 2018) | RGB | 224×224 | 8 | 93.2 | 86.6 |
| DMCL (Garcia et al., 2021) | RGB | 224×224 | 8 | – | 83.6 |
| Debnath *et al.* (Debnath et al., 2021) | RGB | – | – | – | 87.2 |
| FSA-CNN (Jang et al., 2020) | RGB | – | – | 92.2 | 88.1 |
| Piergiovanni *et al.* (Piergiovanni & Ryoo, 2021) | RGB | – | – | 93.7 | – |
| ViewCon (Shah et al., 2023b) | RGB | 224×224 | 32 | 98.0 | 91.4 |
| DVANet (Siddiqui et al., 2024) | RGB | – | – | 98.2 | 93.4 |
| π-ViT (Reilly & Das, 2024) | RGB | 224×224 | 16 | 97.9 | 94.0 |
| *Self-Supervised Methods* | | | | | |
| Li *et al.* (Li et al., 2018) | Flow | – | 6 | 83.4 | 80.9 |
| GL-Transformer (Kim et al., 2022) | Pose | – | 300 | 83.8 | 76.3 |
| AimCLR (Guo et al., 2022) | Pose | – | – | 89.2 | 83.0 |
| HaLP (Shah et al., 2023a) | Pose | – | 64 | 88.6 | 82.1 |
| Vyas *et al.* (Vyas et al., 2020) | RGB | 112×112 | – | 86.3 | 82.3 |
| VideoMAE (Tong et al., 2022)† | RGB | 128×128 | 16 | 92.7 | 85.2 |
| ViewCLR (Das & Ryoo, 2023) | RGB | 128×128 | 32 | 94.1 | 89.7 |
| MV2MAE (Ours) | RGB | 128×128 | 16 | **95.9** | **90.0** |

Table 1: Comparison with state-of-the-art on cross-view and cross-subject action recognition benchmarks of NTU-60 dataset. **Top:** supervised methods using multiple modalities (RGB+Pose), and RGB only, **Bottom:** self-supervised methods using any modality. We report top-1 accuracy after finetuning as in (Li et al., 2018; Das & Ryoo, 2023; Vyas et al., 2020; Kim et al., 2022) We perform multi-crop testing with 5 clips and 10 crops for each, following (Das & Ryoo, 2023). †: using their publicly available implementation.

for information of the target viewpoint, we implicitly use the view information from the visible patches of target viewpoint. We also compare with VideoMAE where we use the same amount of data and match the number of optimization steps, and observe consistent gains compared to this single-view baseline. Moreover, it is noteworthy that most of the *supervised methods* (in Table 1, 2, 4, 6) use a resolution of 224×224 or higher, and despite using a much lower resolution of 128×128, MV2MAE shows strong performance.

On the ETRI dataset (Table 3), MV2MAE improves the action classification accuracy by +1.4% compared to (Dokkar et al., 2023). ETRI does not include an official cross-view benchmark, and thus cross-view generalization is not directly evaluated on this dataset.

## 4.4 Transfer Learning Results

Transfer learning is an important setting for evaluating the generalization capabilities of pre-trained models. The model is initialized using pre-trained weights, and fine-tuned on smaller datasets. We perform transfer learning experiments on three action recognition datasets: 1) NUCLA, 2) PKU-MMD-II, and 3) ROCOG-v2.

NUCLA (Wang et al., 2014) is a multi-view action recognition dataset consisting of 1493 videos spanning 10 action classes. Each activity has been captured from three viewpoints, and we follow the cross-view protocol

| Method | Modality | Resolution | # Frames | NTU-120 (%) | |
| --- | --- | --- | --- | --- | --- |
| | | | | xset | xsub |
| *Supervised Methods* | | | | | |
| Hu *et al.* (Hu et al., 2018) | RGB+Depth | – | – | 44.9 | 36.3 |
| Hu *et al.* (Hu et al., 2015) | RGB+Depth | – | – | 54.7 | 50.8 |
| Separable STA (Das et al., 2019) | RGB+Pose | – | 64 | 82.5 | 83.8 |
| VPN (Das et al., 2020) | RGB+Pose | – | 64 | 87.8 | 86.3 |
| VPN++ (Das et al., 2021) | RGB+Pose | – | 64 | 92.5 | 90.7 |
| 3DA (Kim et al., 2023) | RGB+Pose | 224×224 | 12 | 91.4 | 90.5 |
| PoseC3D (Duan et al., 2022) | RGB+Pose | – | 32 | 96.4 | 95.3 |
| PEM (Liu & Yuan, 2018) | Pose | – | – | 66.9 | 64.6 |
| 2s-AGCN (Lei et al., 2019) | Pose | – | 300 | 84.9 | 82.9 |
| MS-G3D Net (Liu et al., 2020b) | Pose | – | 300 | 88.4 | 86.9 |
| CTR-GCN (Chen et al., 2021b) | Pose | – | 64 | 90.6 | 88.9 |
| ProtoGCN (Liu et al., 2025) | Pose | – | – | 92.2 | 90.9 |
| Hyper-GCN (Zhou et al., 2025) | Pose | – | 300 | 92.0 | 90.9 |
| Two-streams (Simonyan & Zisserman, 2014) | RGB | – | – | 54.8 | 58.5 |
| Liu *et al.* (Liu et al., 2020a) | RGB | – | *all* | 54.8 | 58.5 |
| I3D (Carreira & Zisserman, 2017) | RGB | 224×224 | 250 | 80.1 | 77.0 |
| DMCL (Garcia et al., 2021) | RGB | 224×224 | 8 | 84.3 | – |
| ViewCon (Shah et al., 2023b) | RGB | 224×224 | 32 | 87.5 | 85.6 |
| DVANet (Siddiqui et al., 2024) | RGB | – | – | 91.6 | 90.4 |
| $\pi$-ViT (Reilly & Das, 2024) | RGB | 224×224 | 16 | 92.9 | 91.9 |
| *Self-Supervised Methods* | | | | | |
| GL-Transformer (Kim et al., 2022) | Pose | – | 300 | 68.7 | 66.0 |
| AimCLR (Guo et al., 2022) | Pose | – | – | 76.7 | 76.4 |
| HaLP (Shah et al., 2023a) | Pose | – | 64 | 73.1 | 72.6 |
| VideoMAE (Tong et al., 2022)† | RGB | 128×128 | 16 | 82.4 | 79.7 |
| ViewCLR (Das & Ryoo, 2023) | RGB | 128×128 | 32 | 86.2 | 84.5 |
| MV2MAE (Ours) | RGB | 128×128 | 16 | **87.1** | **85.3** |

Table 2: Comparison with state-of-the-art on cross-setup and cross-subject action recognition benchmarks of NTU-120 dataset. **Top:** supervised methods using multiple or single modality, **Bottom:** self-supervised methods using any modality. We report top-1 accuracy after finetuning as in (Das & Ryoo, 2023; Vyas et al., 2020; Kim et al., 2022), *etc.* We perform multi-crop testing with 5 clips and 10 crops for each, following (Das & Ryoo, 2023). †: using their publicly available implementation.

| Method | Modality | Resolution | # Frames | ETRI (%) |
| --- | --- | --- | --- | --- |
| | | | | xsub |
| *Supervised Methods* | | | | |
| ESE-FN (Shu et al., 2022) | RGB+Pose | – | 64 | 95.9 |
| FSA-CNN (Jang et al., 2020) | RGB | – | – | 90.6 |
| ConViViT (Dokkar et al., 2023) | RGB | – | – | 95.1 |
| *Self-Supervised Methods* | | | | |
| VideoMAE (Tong et al., 2022)† | RGB | 128×128 | 16 | 93.4 |
| MV2MAE (Ours) | RGB | 128×128 | 16 | **96.5** |

Table 3: Comparison with state-of-the-art cross-subject action recognition benchmark of ETRI dataset. MV2MAE performs better than prior work, which are all supervised approaches. We perform multi-crop testing with 5 clips and 10 crops for each. †: using their publicly available implementation.

for our experiments. PKU-MMD-II (Liu et al., 2017) is another dataset for 3D action understanding,

| Method | Modality | Resolution | # Frames | NUCLA (%) xview |
|---|---|---|---|---|
| *Supervised Methods* | | | | |
| STA (Das et al., 2019) | RGB+Pose | – | 64 | 92.4 |
| VPN (Das et al., 2020) | RGB+Pose | – | 64 | 93.5 |
| VPN++ (Das et al., 2021) | RGB+Pose | – | 64 | 93.5 |
| DA-Net (Wang et al., 2018) | RGB | – | – | 86.5 |
| Glimpse Cloud (Baradel et al., 2018) | RGB | 224×224 | 8 | 90.1 |
| I3D (Carreira & Zisserman, 2017) | RGB | 224×224 | 250 | 88.8 |
| ViewCon (Shah et al., 2023b) | RGB | 224×224 | 32 | 91.7 |
| DVANet (Siddiqui et al., 2024) | RGB | – | – | 96.5 |
| Hyper-GCN (Zhou et al., 2025) | Pose | – | 300 | 97.6 |
| *Self-Supervised Methods* | | | | |
| MS$^2$L (Lin et al., 2020) | Pose | – | 200 | 86.8 |
| Li *et al.* (Li et al., 2018) | Depth | – | 6 | 62.5 |
| Colorization (Yang et al., 2021) | Depth | – | 50 | 94.0 |
| Vyas *et al.* (Vyas et al., 2020) | RGB | 112×112 | – | 83.1 |
| ViewCLR (Das & Ryoo, 2023) | RGB | 128×128 | 32 | 89.1 |
| MV2MAE (Ours) | RGB | 128×128 | 16 | **97.6** |

Table 4: Transfer learning on NUCLA. Self-supervised methods are pre-trained on NTU-60 dataset. MV2MAE significantly outperforms other methods showing remarkable transfer capability of our representations.

| Method | Modality | Resolution | # Frames | PKU-MMD (%) xsub |
|---|---|---|---|---|
| *Self-Supervised Methods* | | | | |
| CrosSCLR-B (Zolfaghari et al., 2021) | Pose | – | – | 52.8 |
| CMD (Mao et al., 2022) | Pose | – | 64 | 57.0 |
| HaLP (Shah et al., 2023a) | Pose | – | 64 | 57.3 |
| MV2MAE (Ours) | RGB | 128×128 | 16 | **60.1** |

Table 5: Transfer learning on PKU-MMD. All methods use NTU-120 dataset for pre-training. MV2MAE surpasses other unsupervised methods, all of which use Pose modality.

consisting of 6945 videos from 51 activity classes. Following prior work (Shah et al., 2023a), we use the phase 2 of the dataset and evaluate our approach on the cross-subject setting. ROCOG-v2 (Reddy et al., 2023) is a gesture recognition dataset consisting of 304 ground viewpoint videos from 7 gestures.

As shown in Table 4, our method achieves better performance than prior *supervised and unsupervised* methods on the NUCLA dataset. MV2MAE improves the action classification accuracy by +8.5% upon the previous RGB-based SOTA SSL approach (Das & Ryoo, 2023) and by +1.1% compared to best *supervised* pre-trained approach (Siddiqui et al., 2024). On the PKU-MMD-II dataset (Table 5), our method shows an improvement of +2.8% compared to prior work, all of which are based on Pose modality. Finally, we show results on the ROCOG-v2 dataset in Table 6, where we gain an improvement of +2.0%.

These transfer learning results clearly demonstrates that the representations learnt using our approach generalize well. It is noteworthy that although the Pose modality shows superior performance in supervised setting (Table 2), it lags behind when used for self-supervised learning in both in-domain fine-tuning (Table 2) and transfer learning (Table 4 and 5) settings.

| Method | Modality | Resolution | # Frames | **ROCOG (%)** |
|---|---|---|---|---|
| Reddy *et al.* (Reddy et al., 2023) | RGB | 256×256 | 16 | 87.0 |
| MV2MAE (Ours) | RGB | 128×128 | 16 | **89.0** |

Table 6: Transfer learning on ROCOG ground dataset.

### 4.5 Synthetic Multi-View Data for Pre-Training

Real multi-view videos can be difficult to acquire and can pose privacy concerns. As an alternative, we investigate the use of synthetic multi-view action recognition data. In our experiments, we pre-train models using synthetic data from SynADL (Hwang et al., 2021) dataset and fine-tune and evaluate on real data from ETRI (Jang et al., 2020) dataset. We compare synthetic pre-training (green) with real pre-training (orange). We observe that if the amount of synthetic data used is same (1x) as the amount of real data, there is a performance drop due to the domain difference. However, increasing the amount of synthetic data used for pre-training allows synthetic pre-training to surpass real pre-training, as seen in Figure 3. These results suggest

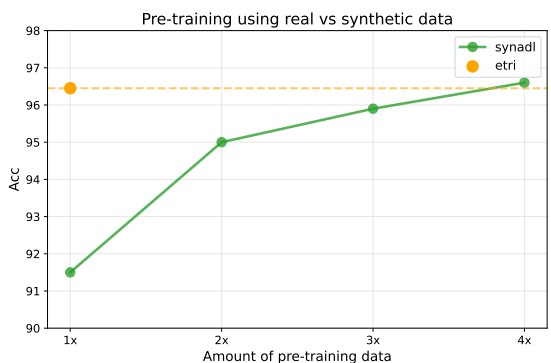

Figure 3: **Pre-training using synthetic data.** Pre-training using more (4x) synthetic data beats pre-training using real data on the same real test set.

that synthetic data can serve as a scalable and effective alternative for pre-training.

### 4.6 Ablation Study and Analysis

We perform ablation experiments on the cross-subject benchmark of the NTU-120 dataset. For ablations regarding viewpoints (Table 10 and 11), we use the ETRI dataset as it contains four synchronised viewpoints compared to NTU-120 which has three.

**How much emphasis to place on reconstructing moving patches?** The motion weights in MV2MAE can be adjusted to modulate the emphasis on moving patches, using the temperature parameter. As shown in Figure 2, a lower temperature places more focus on reconstructing patches with more motion, and increasing the temperature increases the weight given to the background pixels. Figure 4 shows the influence of the temperature parameter on accuracy. From the plot, we see that a temperature value of 60 performs best, which is used in all our experiments.

Increasing the weights of background patches by increasing temperature degrades the performance. This is because it is trivial to reconstruct the background patches by copy-paste from nearby frames. The performance degrades significantly to 82.46% if each patch in the video is weighted equally, since the the number of static pixels are more than those with motion. Note that these experiments are trained for 1200 epochs.

**Masking Ratio.** We study the impact of masking ratio in Table 7. Note that the optimal masking ratio is lower in our multi-view setting than the standard MAE (Tong et al., 2022), which we hypothesize

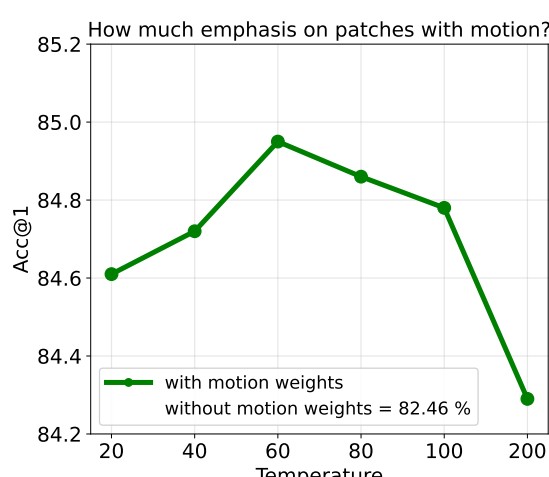

Figure 4: **Temperature parameter** $\tau$ of motion weights modulates the focus on static vs moving re-

is because more information is needed from individual views to effectively infer cross-view geometry.

**Model Scaling.** To study how the performance scales with models of different capacities, we pre-train using ViT-T, ViT-S, and ViT-B encoders and compare the fine-tuning accuracy on the NTU-120 cross-subject setting in Table 8. MV2MAE can effectively pre-train larger models using the same amount of data. All model scaling experiments perform pre-training for 800 epochs as opposed to the default 1600 due to computational constraints.

| Masking ratio ($\rho$) | 0.6 | 0.7 | 0.8 | 0.9 |
|---|---|---|---|---|
| NTU-120 xsub (%) | 83.6 | **85.3** | 84.3 | 83.4 |

Table 7: **Masking Ratio.** MV2MAE performs best with a masking ratio of 0.7. Default in gray .

| Backbone | ViT-T | ViT-S | ViT-B |
|---|---|---|---|
| NTU-120 xsub (%) | 82.0 | 83.4 | 85.1 |

Table 8: **Model Capacity.** Our approach scales effectively with bigger models. Default in gray .

**Ablation of Cross-view Decoder.** The cross-view decoder significantly boosts performance in both xsub and xset settings, with more improvement on the xview benchmark (+2.9%) than on xsub benchmark (+2.2%), showing effectiveness when evaluating on unknown viewpoints. In the experiment without cross-view decoder, the model is trained using single-viewpoint videos ignoring synchronization, on the same amount of data.

| Cross-View Decoder | NTU-120 xsub (%) | NTU-120 xset (%) |
|---|---|---|
| ✗ | 83.1 | 84.2 |
| ✓ | 85.3 (+2.2) | 87.1 (+2.9) |

Table 9: **Cross-View Decoder** leads to substantial improvements on both xview and xset benchmarks.

**Visualizing Cross-Attention Maps and Reconstructions.** Here, we analyze the cross-view decoder by visualizing the cross-attention maps and the cross-view reconstruction quality. The cross-attention maps are visualized in Figure 5. The first and second rows show the input and masked input frames from the target viewpoint, with a masked query token circled in red. The third row shows the reconstructed target view from the cross-view decoder. The last row shows the cross-attention map corresponding to the query token overlaid on the source view frames. We can see that model is able to find matching regions in the source viewpoint, demonstrating the learnt geometry.

**How many source views to use?** For the cross-view decoder, we study the effect of number of source viewpoints used in Table 10. For these experiments, all viewpoints used are chosen randomly from available synchronized views. The performance is similar when using one or two source viewpoints. We observe that the fine-tuning accuracy drops if we use more source viewpoints for reconstructing the target viewpoint, by making the reconstruction task easier.

| # Source Views | 1 | 2 | 3 |
|---|---|---|---|
| ETRI xsub (%) | 94.0 | 93.9 | 93.1 |

Table 10: **Number of Source Views.** Using more source views makes reconstruction task easier and degrades performance. Default in gray .

**How different should the views be?** A natural question that arises when creating such datasets is which viewpoint to capture? Given a target viewpoint, we study how far should the source view be. We study this by fixing the target view to be View1 (shown in Figure 6), and varying the source view to be View2, View3 or View4 . The results are reported in Table 11, which shows that the performance drops slightly if the chosen target and source viewpoints are separated by a lot. For all experiments in the paper, we do not fix the target view, and choose source and target views randomly for added diversity.

**Single-Clip Single-Crop Inference** We compare single-clip, single-crop setting to the full multi-crop evaluation with 5 clips and 10 crops from each in the cross-subject setting of NTU-60 and NTU-120 datasets.

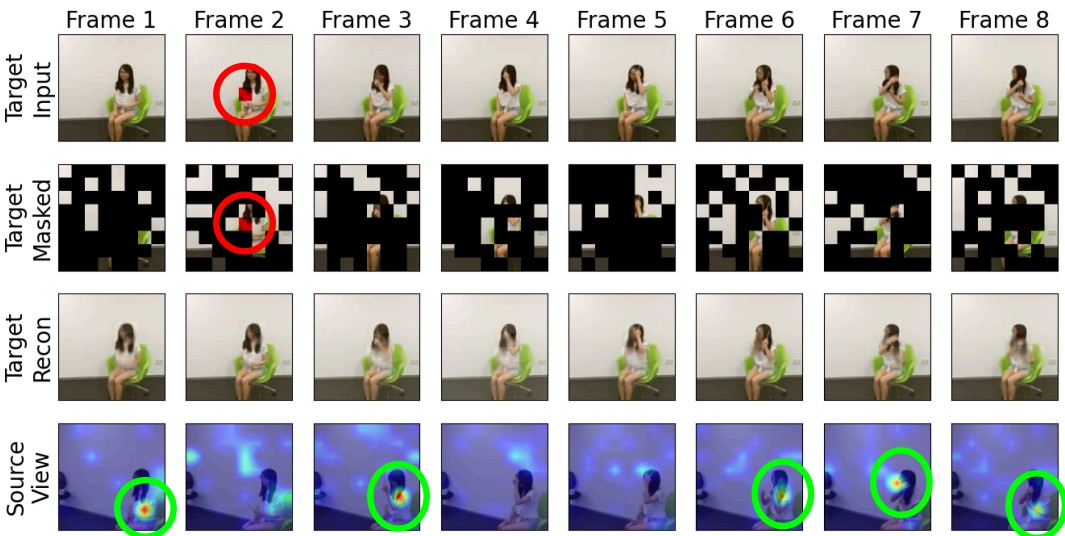

Figure 5: **Cross-View Decoder Qualitative Analysis.** We visualize the reconstructions and cross-attention maps from the cross-view decoder. **First row:** Target viewpoint input frames, **Second row:** Masked input frames from target viewpoint, **Third row:** Reconstruction of target view from the cross-view decoder, **Last row:** Cross-attention maps visualized on source view frames. The red circle indicates a query token in the target viewpoint whose attention maps are visualized in the last row. Green circles shows that the model is able find and leverage matching regions across viewpoints for cross-view reconstruction.

On NTU-60, the performance drops from 90.0% to 82.1%, and on NTU-120, the performance drops from 85.3% to 77.3%.

Figure 6: Example of synced views from ETRI dataset.

| Source View | View2 | View3 | View4 |
|---|---|---|---|
| ETRI xsub (%) | 94.7 | 94.4 | 94.3 |

Table 11: **Which views to choose?** View2 is the closest and View4 the farthest from the target. Performance degrades slightly if the two views are very different.

## 5 Limitations

While MV2MAE demonstrates strong performance on controlled datasets with static backgrounds (e.g., NTU, ETRI), its efficacy on unconstrained, in-the-wild multi-view videos with complex camera motions and dynamic backgrounds (e.g., EgoExo4D (Grauman et al., 2024), LEMMA (Jia et al., 2020)) remains to be fully explored. To address this, future work could investigate pre-training on larger-scale datasets featuring more realistic motions, or explore robustness strategies such as artificially introducing background motion during fine-tuning.

## 6 Broader Impact

While multi-view action recognition advances fields like elderly care, its potential application in surveillance raises privacy concerns regarding the capture of detailed geometric information. To mitigate these risks, we investigate synthetic data as a scalable, privacy-preserving alternative for pre-training. Although scaling synthetic data allows models to surpass those trained on real-world footage, further research into domain

adaptation is essential to better bridge the synthetic-to-real gap, offering a viable path to minimize reliance on sensitive data without compromising robustness.

## 7 Conclusion

We propose a self-supervised learning approach for harnessing the power of multi-view videos within the masked autoencoder framework. Our method integrates a cross-view reconstruction task, leveraging a dedicated decoder equipped with cross-attention mechanism to instill geometry information into the model. The introduction of a motion-focused reconstruction loss further enhances temporal modeling. We empirically show that our SSL method enables learning robust generalizable multi-view features contributing to better performance when used for full fine-tuning and transfer learning.

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

## A  PyTorch code for motion weights

Given the input video frames, patch-size and the temperature parameter, we provide PyTorch code below for motion weights based on frame difference used in the paper.

**Algorithm 1:** PyTorch code for motion weights.

```
# frames    : input frames of shape [B,C,T,H,W]
# patch_size: (p_time, p_height, p_width)
# t         : temperature parameter

fdiff = frames[:,:,1:,:,:] - frames[:,:,:-1,:,:]
fdiff = torch.cat([fdiff[:,:,0:1,:,:], fdiff], dim=2)
fdiff = rearrange(fdiff,'b c (t p0) (h p1) (w p2) -> b (t h w) (p0 p1 p2 c)', p0=patch_size[0], p1=patch_size[1],
    p2=patch_size[2])
fdiff = torch.abs(fdiff)
fdiff = torch.linalg.vector_norm(fdiff, dim=2, keepdim=True) # B N 1
motion_weights = torch.nn.functional.softmax(fdiff/t, dim=1) # B N 1
```

## B  Architecture Details

The detailed asymmetric architecture of the encoder and decoders is shown in Table 12, Table 13 and Table 14. We have two decoders in our architecture: 1) self-view decoder and 2) cross-view decoder. The self-view decoder only uses self-attention to reconstruct same view whereas the cross-view decoder uses cross-attention in addition to self-attention for reconstructing target viewpoint while also using source viewpoint. These decoders are discarded during fine-tuning. We use 16 frame input and choose ViT-S/16 as our default encoder. We adopt the joint space-time attention (Arnab et al., 2021) for the encoder.

| Stage | Vision Transformer (Small) | Output Sizes |
|---|---|---|
| data | stride $4 \times 1 \times 1$ | $3 \times 16 \times 128 \times 128$ |
| cube | $2 \times 16 \times 16$, $384$ 
 stride $2 \times 16 \times 16$ | $384 \times 8 \times 64$ |
| mask | random mask 
 *mask ratio* $= \rho$ | $384 \times 8 \times [64 \times (1\text{-}\rho)]$ |
| encoder | $\begin{bmatrix} \text{MHA}(384) \\ \text{MLP}(1536) \end{bmatrix} \times 12$ | $384 \times 8 \times [64 \times (1\text{-}\rho)]$ |
| projector | MLP($192$) & 
 *concat learnable tokens* | $192 \times 8 \times 64$ |

Table 12: **Encoder of MV2MAE.** The encoder processes 16-frame input clips from source and target views, and the encoded representations of the visible tokens are combined with the learnable mask tokens, before passing through the decoder.

## C  Additional Implementation Details

The pre-training and fine-tuning hyper-parameter settings for NTU-60, NTU-120 and ETRI datasets are given in Table 15 and Table 16.

## D  Single-View vs Multi-View Inference

At test time, multiple viewpoints of an activity are available in the cross-subject setting. However, evaluation in prior work is carried out using single-view at a time, following the original benchmark (Shahroudy et al.,

| Stage | Transformer | | Output Sizes |
|---|---|---|---|
| self-view decoder | $\begin{bmatrix} \text{MHA}(192) \\ \text{MLP}(768) \end{bmatrix}$ | ×4 | 192×**8**×64 |
| projector | MLP(1536) | | 1536×**8**×64 |
| reshape | *from* 1536 *to* 3×2×16×16 | | 3×16×128×128 |

Table 13: **Self-view decoder of MV2MAE.** It takes the source and target view tokens and reconstructs both the views independently.

| Stage | Transformer | | Output Sizes |
|---|---|---|---|
| cross-view decoder | $\begin{bmatrix} \text{MHCA}(192) \\ \text{MHA}(192) \\ \text{MLP}(768) \end{bmatrix}$ | ×4 | 192×**8**×64 |
| projector | MLP(1536) | | 1536×**8**×64 |
| reshape | *from* 1536 *to* 3×2×16×16 | | 3×16×128×128 |

Table 14: **Cross-view decoder of MV2MAE.** The cross-view decoder uses the visible tokens from the source view to reconstruct the missing patches in the target view.

| config | NTU60 | NTU120 | ETRI |
|---|---|---|---|
| optimizer | | AdamW | |
| base learning rate | | 1e-3 | |
| weight decay | | 0.05 | |
| optimizer momentum | | $\beta_1, \beta_2{=}0.9, 0.95$ | |
| batch size | | 1024 | |
| learning rate schedule | | cosine decay | |
| warmup epochs | 320 | 160 | 160 |
| total epochs | 3200 | 1600 | 1600 |
| augmentation | | MultiScaleCrop | |

Table 15: **Pre-training setting.**

| config | NTU60 | NTU120 | ETRI |
|---|---|---|---|
| optimizer | | AdamW | |
| base learning rate | | 1e-3 | |
| weight decay | | 0.1 | |
| optimizer | | $\beta_1, \beta_2{=}0.9, 0.999$ | |
| momentum | | | |
| batch size | | 1024 | |
| learning rate | | cosine decay | |
| schedule | | | |
| warmup epochs | 5 | 10 | 10 |
| training epochs | 35 | 120 | 120 |
| repeated | | 6 | |
| augmentation | | | |
| flip augmentation | | *yes* | |
| RandAug | | (7, 0.5) | |
| label smoothing | | 0.1 | |
| drop path | | 0.1 | |
| layer-wise lr decay | | 0.9 | |

Table 16: **End-to-end fine-tuning setting.**

2016). Though in most practical scenarios, it would be natural to combine the predictions from available synchronized viewpoints for a given activity. We show this comparison of single-view and multi-view inference in Table 17. For multi-view inference, the predictions are combined using late fusion strategy.

Table 17: SV vs MV inference. We perform late fusion for multi-view inference.

| | Cross-Subject (%) | |
|---|---|---|
| Method | NTU-60 | NTU-120 |
| Single-View Inference | 90.0 | 85.3 |
| Multi-View Inference | 91.9 | 87.9 |

# E   Parameter Count Comparison

| Method | Modality | Backbone | #Params (M) |
|---|---|---|---|
| MV2MAE (Ours) | RGB | ViT-S/16 (encoder only) | 22.1 |
| VideoMAE | RGB | ViT-S/16 | 22.1 |
| VideoMAE | RGB | ViT-B/16 | 86.6 |
| VideoMAE | RGB | ViT-L/16 | 304.4 |
| Vyas *et. al.* | RGB | 3D Conv + Conv-LSTM | 72 |
| ViewCLR | RGB | S3D (encoder only) | 9 |
| 2s-AGCN | Pose | 2s-AGCN | 6.9 |
| MS-G3D | Pose | MS-G3D | 2.7–3.2 |
| CTR-GCN | Pose | CTR-GCN | 1.4 |
| AimCLR | Pose | AimCLR | 0.85 |

Table 18: Model parameter counts for MV2MAE and baselines reported in our main comparisons. For MV2MAE, we report encoder-only parameters since decoders are discarded at fine-tuning. For VideoMAE, parameter counts depend on the backbone size used.

