# OpenReview forum: "MV2MAE: Self-Supervised Video Pre-Training with Motion-Aware Multi-View Masked Autoencoders"
_TMLR — Accepted by TMLR_

### Review · Reviewer_eiaw · 2025-10-27

**Summary Of Contributions:**

This paper introduces a self-supervised method for viewpoint-agnostic video representation learning, extending VideoMAE’s masked autoencoding framework. The approach adds (i) a cross-view decoder that reconstructs pixels in one camera view from features derived from a time-synchronized multi-view input, and (ii) a motion-aware, pixel-wise loss that weights reconstruction by estimated motion across frames. Ablations demonstrate that both cross-view reconstruction and motion-aware weighting contribute significantly to performance. The method is validated on three datasets and further transfers to two additional datasets, showing consistent, promising gains for downstream adoption.

**Audience:**

Yes

**Audience Explanation:**

The task of learning viewpoint-agnostic video representations is challenging. Many existing methods rely on pretraining with large-scale Internet videos, assuming ample viewpoint diversity; however, this assumption often fails in indoor settings with limited camera perspectives. Self-supervised approaches that exploit synchronized multi-view data provide a principled alternative: they learn viewpoint-agnostic features directly and mitigate a key limitation of current neural networks, whose operations are not inherently viewpoint-invariant.

**Broader Impact Concerns:**

There is no broader impact section in this paper. I think the study could be leveraged by the vision community to leverage multiview data, perhaps could be explored for ego-exo learning as well. I don't see any ethical concerns in this paper, so having no broader impact section should work fine.

**Claims And Evidence:**

Yes

**Claims Explanation:**

The claims made in this paper are well supported with experimental validation on sufficient number of datasets.

**Requested Changes:**

1. The paper does not discuss several recent works. For example, CrossMAE (Guo et al., CVPR 2024) is a self-supervised method whose decoder uses cross-attention resembling the cross-view decoder proposed here. RGB viewpoint agnostic method - 3D TRL (Shang et al., NeurIPS 2022).

2. Relevance of Table 3 / Section 4.5 - The SynADL model is pre-trained on ETRI and evaluated on ETRI, i.e., same-domain. A more informative setup would select common actions from other datasets (e.g., NTU RGB+D, Toyota Smarthome (Das et al., ICCV 2019)) and perform action recognition there. The baseline should be a model trained only on single-view data from the target dataset. This would better demonstrate the value of the method and of multi-view data, even for synthetic data.

3. VideoMAE comparisons are quite important in this paper. Only a single result is presented and not in a table. Please include a full table reporting results across all datasets.

4. Indicate that decoders shown in the same color share weights to avoid reader confusion in Figure 1.

5. Another baseline could have been a VideoMAE trained on Kinetics-400. For fairness, randomly sub-sample Kinetics to match the sample count of NTU RGB+D. This directly tests whether multi-view data is necessary, addressing the counterargument that web-scale data already captures diverse viewpoints; if not, it strengthens the case for MV2MAE.

---

> ### Author Response · Authors · 2026-01-03
> **Response to Reviewer eiaw**
>
> We thank the reviewer for their constructive feedback and for recognizing our method’s validation on multiple datasets and its promise for downstream adoption. We have revised the manuscript to address your comments.
>
> 1. We agree that discussing these works clarifies the positioning of our method. We have updated Section 2.2 to include CrossMAE and 3D TRL.
>
> 2. We want to clarify that in Section 4.5, we currently demonstrate a Synthetic-to-Real transfer protocol by pre-training on SynADL (synthetic) and fine-tuning on ETRI (real), and emphasize that there is a significant domain gap between the synthetic and real data. The single-view baseline is in the ablation provided in Table 9, where the results without cross-view decoder are effectively models trained on single-viewpoint data.
>
> 3. We appreciate the feedback and we have now added VideoMAE results for all the main evaluations in Table 1, 2, 3.
>
> 4. The revised manuscript reflects this change in the caption of Figure 1.
>
> 5. Although Kinetics-400 is large-scale and diverse in content, previous work on multi-view action recognition has noted that it is not viewpoint-balanced and is dominated by front-facing views. It has been shown in previous work (e.g. ViewCLR) that Kinetics-400 pre-trained models hence do not perform well especially under cross-view evaluation setting.

---

### Review · Reviewer_sUVt · 2025-11-07

**Summary Of Contributions:**

The paper introduces MV2MAE, a self-supervised pre-training strategy designed for synchronized multi-view videos. The method builds upon the MAE paradigm and makes two primary contributions: 1) a cross-view reconstruction task where a cross-attention decoder predicts a target viewpoint’s masked patches from visible source-view tokens to force the encoder to learn view-invariant features; 2) a motion-weighted loss that emphasizes reconstruction of dynamic (high motion) regions to reduce trivial background reconstruction and improve temporal understanding. Their method achieves SOTA  performance among other self-supervised approaches on NTU60, NTU120, and ETRI, and also demonstrates strong transfer learning gains on smaller datasets such as NUCLA, PKU-MMD, ROCOG. As multi-view videos can be difficult to obtain, the authors perform a preliminary study on synthetic multi-view data as a scalable alternative.

**Additional Comments:**

Clarifying question: in Figure 4, why does the result at $\tau=60$ not match the result in Table 2 for xsub (85.3)? It seems like this ablation might have been trained for less epochs (similar to the Model Scaling analysis), which is fine but isn’t mentioned

**Audience:**

Yes

**Audience Explanation:**

This work addresses the task of self-supervised learning in videos in the case where multiple time-synchronized views are available, and is likely to interest readers working on tasks such as multi-view representation learning and cross-view knowledge transfer. Given the requirement of multiple synchronized views, it can also be of interest to readers working on video surveillance with security cameras, as multiple cameras often monitor the same scene.

**Claims And Evidence:**

Yes

**Claims Explanation:**

The results support the claims made in the submission. On NTU60, MV2MAE surpasses ViewCLR (+1.8 points on xview, +0.3 on xsub), and on NTU120 (+0.9 on view, +1.2 on sub). Accuracy on the ETRI dataset surpasses ConViViT by +1.4. The transferability of the method is validated on NUCLA, PKU, and ROCOG, where MV2MAE outperforms representative baselines. The effect of each contribution is validated through ablation studies: removing the cross-view decoder reduces NTU120 accuracy by 2-3%, equal weighting instead of motion weighting reduces accuracy by ~2.5%. Additional analysis (masking ratio, temperature sensitivity, source-view count) validate the design choices of their method.

**Requested Changes:**

The reviewer would like to see how MV2MAE performs on videos with more complex and realistic motions, or for the authors to provide a limitations section regarding this. The datasets explored in this paper are in very tightly controlled environments: the background scenes are static, the actors are always centered in the frames, and the motions that define each action is emphasized. This is not the case for real-world multi-view data (datasets such as LEMMA, EgoExo4D, Toyota Smarthome), which may limit the success of the proposed method.

From Figure 4, MV2MAE seems highly sensitive to the motion-weighted loss as without it the performance drops to 82%. The current method used to compute motion is frame differencing, which suggests that the method likely fails if videos contain motion unrelated to the action being performed (such as outdoor scenes, or homes with windows or a TV). Can the authors examine how their method performs on videos containing background motion?

Can the authors present the action classes where MV2MAE improves performance? This can make the paper more convincing if the actions being improved are highly dependent on understanding motions

---

> ### Author Response · Authors · 2026-01-03
> **Response to Reviewer sUVt**
>
> We thank the reviewer for their encouraging comments and for recognizing the value of our contributions. We appreciate the thoughtful feedback.
>
> We have added a *Limitations* section in the revised manuscript to discuss the bias in the datasets used and the need for improved algorithms and data for tackling more complex real-world scenarios. To address this, future work could investigate pre-training on larger-scale datasets featuring more realistic motions, or explore robustness strategies such as artificially introducing background motion during fine-tuning.
>
> We acknowledge that our method would likely be sensitive to unrelated background motion in its current form. Since the datasets used for pre-training and fine-tuning (e.g., NTU, ETRI) feature largely static environments, videos with active backgrounds represent an out-of-domain scenario where the model would incorrectly prioritize irrelevant noise. To address this "in-the-wild" challenge, we would need to either close the domain gap by incorporating real or synthetically composited dynamic backgrounds into the training data, or refine the motion-weighted reconstruction loss $\mathcal{L}_m$ to utilize more semantic motion priors that can distinguish human action from background clutter.
>
> The top classes where MV2MAE improves the performance are:
> - Wear shoe / Take off shoe
> - Pick up / Drop
> - Handshake / Exchange things with other person
>
> Thank you for the observation regarding the temperature ablation in Figure 4. The discrepancy is due to the models for ablation being trained for fewer epochs (1200). We have updated the paper to reflect this.

---

### Review · Reviewer_PaoQ · 2025-12-17

**Summary Of Contributions:**

This paper proposes MV2MAE, a masked-autoencoder-style pretraining approach fpr synchronized multi-view videos. The core idea is to augment standard masked video modeling with a a dedicated cross-view decoder uses cross-attention from a masked target view to visible tokens from a source view, encouraging representations that capture geometric/viewpoint relationships. In addition, the paper introduces a simple motion-weighted reconstruction loss to reduce the dominance of easily reconstructed static/background regions. The overall pretraining objective combines self-view reconstruction on both views plus cross-view reconstruction in both directions.
The authors evaluate on standard multi-view action recognition benchmarks (NTU-60/120, ETRI) and transfer settings (e.g., NUCLA/PKU-MMD/ROCOG). They report strong performance among SSL methods; e.g., on NTU-60 they report **95.9% (xview)** and **90.0% (xsub)** with RGB at 128×128 and 16 frames. Ablations show that the cross-view decoder is a meaningful contributor. The paper also includes a synthetic-data study suggesting that sufficiently scaling synthetic multi-view data can match or surpass real-data pretraining, but still requiring much more sinthetic samples.

**Additional Comments:**

You might also briefly mention adjacent multi-view masked modeling work outside action recognition (e.g., multi-view masked autoencoders/world models used in robotics) to contextualize the broader relevance [Seo et. al.](https://proceedings.mlr.press/v202/seo23a/seo23a.pdf).

**Audience:**

Yes

**Audience Explanation:**

Multi-view video is common in real deployments (multi-camera rooms, assisted living, robotics, sports), and learning viewpoint-robust representations without requiring pose/depth is a relevant problem. The approach is appealing because it is simple and modular: it stays close to the MAE paradigm while injecting cross-view structure via a cross-attention decoder, and discards decoders at fine-tuning time (so downstream cost remains similar to a standard encoder). The inclusion of synthetic multi-view pretraining analysis also makes the work interesting for audiences thinking about privacy, scalability, and data sourcing.

**Broader Impact Concerns:**

- Multi-view action recognition methods can be applied to surveillance or monitoring scenarios. Since the work is explicitly about leveraging synchronized multi-camera video, it can amplify capabilities in settings where privacy concerns are significant.

I suggest adding a short discussion on: (i) privacy considerations for collecting synchronized multi-view data, and (ii) whether synthetic multi-view generation (simulation, rendering, etc.) could be a safer alternative and what tradeoffs remain.

**Claims And Evidence:**

Yes

**Claims Explanation:**

The main algorithmic claims are supported by (i) a clear method description, (ii) multiple benchmark evaluations, and (iii) targeted ablations.

-  Cross-view reconstruction is clearly defined and motivated (learning viewpoint-robust representations by requiring cross-view extrapolation), and the architectural mechanism is described concretely.

-  The full pretraining loss is explicit and includes both self-view and cross-view terms, making it easy to understand what is optimized.

- Ablations support key components: enabling the cross-view decoder yields consistent improvements across NTU-120 splits.The motion-weighting temperature study indicates an optimum around \tau=60 and shows a notable drop when weights become uniform.

- Experimental protocol details are present (e.g., 16 RGB frames with stride 4; 128×128 input; ViT-S/16; 1600 epochs), and evaluation uses 5 temporal clips x 10 crops.

That said, a few aspects weaken how clear the evidence is for some of the paper’s comparative and "practicality" claims:

- Some comparisons mix backbones, resolutions, and evaluation settings; the paper notes supervised methods often use ≥224 resolution while MV2MAE uses 128, but the main tables still leave room for ambiguity about fairness (e.g., crops/clips, compute, model size).

- The paper states MV2MAE is faster/more memory efficient than ViewCLR because ViewCLR uses MoCo-style components, but it would be more convincing with a simple quantitative comparison (GPU memory, training throughput).

Overall, the evidence supports the core claims, but I think the paper would benefit from clarifications and a couple of additional baselines to make the comparisons more rigorous.

**Requested Changes:**

### Major / high-priority

1.  **Add/clarify a “single-view MAE” baseline (compute-matched)**
    -   You already include an ablation disabling the cross-view decoder (Table 9), which is helpful, but it is still a multi-view training regime (two synchronized views available).
    -   Please add an explicit baseline where you ignore synchronization and pretrain a standard VideoMAE/MAE on the same underlying videos (treat each view independently as its own sample), matched for total compute (e.g., same number of tokens/updates). This would directly answer: Do we need cross-view, or do gains come mostly from "more data/augmentation" by having multiple views?


2.  **Make evaluation protocol explicit in each main results table (clips/crops)**

    -   The paper states test-time evaluation uses 5 temporal clips and 10 crops, which is a substantial inference-time budget.

    -   Please add a column/footnote in the main result tables specifying:
        -   number of temporal clips
        -   number of spatial crops
        -   whether results are single-crop vs multi-crop

    -   If prior work uses different evaluation budgets, consider adding a “single-clip, single-crop” evaluation for MV2MAE (and maybe ViewCLR if feasible) to contextualize inference cost.

3.  **Clarify backbone choice and comparison fairness (ViT-S, params/FLOPs)**

    -   The default encoder is ViT-S/16. Please clarify whether the main SSL baselines you compare against use the same backbone capacity, or if not, include parameter count and/or FLOPs for each method (or at least for the key SSL baselines).
    -   You mention parameter counts for one comparison (e.g., Vyas et al. uses more parameters); extending this systematically would strengthen the SOTA claim.

4.  **Explain and/or ablate the 128×128 resolution choice**

    -   You state you downsample to 128×128 following ViewCLR. This is reasonable, but many ViT/video works commonly use ≥224.
    -   Please either:
        -   justify the choice in terms of compute/memory constraints (especially since pretraining uses two synchronized views), and/or
        -   provide a small ablation showing how performance scales with resolution (e.g., 128 vs 160/224) for at least one dataset.

5.  **Clarify viewpoint geometry and train/test camera relationship (esp. NTU)**

    -   The paper states NTU cross-view trains on cameras 2&3 and tests on camera 1, but it would help readers to know how those cameras are positioned.

    -   In the NTU dataset documentation, the three cameras correspond to horizontal angles −45°, 0°, +45°, and “camera 1” corresponds to the 45° views while cameras 2/3 capture front/side views. Adding a one-sentence clarification/diagram would prevent confusion (e.g., whether the test camera is "far" or "intermediate" relative to training cameras).

    -   You already do a nice viewpoint-distance analysis on ETRI (View2 closest vs View4 farthest) with only slight degradation when views are very different. A similar clarification for NTU would align with this analysis.

6.  **ETRI: explain “cross-subject only” and consider adding a cross-view protocol**

    -   The paper says ETRI is evaluated using the cross-subject benchmark from the dataset paper, which likely explains why there is no cross-view result.

    -   Still, since ETRI has 8 synchronized viewpoints, it would be valuable to either:
        -   define a simple cross-view split (train on a subset of cameras, test on held-out cameras), or
        -   explicitly discuss why that is not feasible/standard, and how your method would be expected to behave.

7.  **Deepen the analysis of masking ratio vs number of views (“information leakage”)**

    -   You find the optimal masking ratio is lower (0.7) than in standard VideoMAE-style settings (around ~0.9) and hypothesize this is because more info is needed per view to infer geometry.

    -   You also show that using more source views can make reconstruction too easy and hurts downstream accuracy.

    -   I suggest adding one compact experiment/plot: accuracy as a function of (number of source views, masking ratio), to show the tradeoff between task difficulty and representation quality. This would make the “view leakage” story much more concrete.


### Minor / presentation and clarity

1.  In the compiled PDF, Figures 3 and 4 appear rasterized/blurry. Please export plots as vector (PDF) or increase resolution so the paper is readable when zoomed.

2. In the viewpoint-distance discussion, the text references “Table 7” while the view-distance result is says Figure 7; please correct this.

3. Since you argue ViewCLR is heavier due to MoCo queues and double networks, a small table with peak GPU memory and iterations/sec (or GPU-days) would substantiate this nicely.

---

> ### Author Response · Authors · 2026-01-03
> **Response to Reviewer PaoQ**
>
> We thank the reviewer for their detailed and constructive feedback, and for acknowledging the strong performance and modularity of our proposed MV2MAE approach. We appreciate the suggestion to clarify our baselines and experimental protocols, which we believe has significantly strengthened the manuscript. We have addressed the specific concerns below.
>
> **1. Add/clarify a “single-view MAE” baseline (compute-matched)**
> - We would like to clarify that the ablation study in Table 9 (Row 1: "Cross-View Decoder: X") represents exactly this baseline. In this experiment, the model is trained without the cross-view decoder and without the cross-view reconstruction task. It treats the videos effectively as single-view samples (ignoring synchronization), but still utilizes the motion-weighted reconstruction loss.
> - We have revised Section 4.6 to indicate this.
>
> **2. Make evaluation protocol explicit in each main results table (clips/crops)**
> - We have added a footnote in each table explicitly stating our testing protocol (5 temporal clips × 10 spatial crops), which follows the protocol used in ViewCLR (Das & Ryoo, 2023)
> - We adhered to the multi-crop setting as all the SSL approaches using RGB modality evaluate using this protocol. Moreover, the code for ViewCLR is not publicly available, preventing us from fair comparison in a single-clip, single-crop setting.
>
> **3. Clarify backbone choice and comparison fairness (ViT-S, params/FLOPs)**
> - MV2MAE uses a ViT-S/16 backbone with ~22M parameters. Among the other RGB-based SSL approaches we compare with, Vyas et. al. 2020, uses ~72M parameters. Parameter count for other methods are not available.
>
> **4. Explain and/or ablate the 128×128 resolution choice**
> - Our resolution choice is primarily motivated by computational and memory constraints in processing synchronized multi-view videos. Since our pre-training objective involves processing two concurrent video clips (source and target views), the number of tokens doubles compared to standard single-view training.
> - This choice also allows us to fairly compare with ViewCLR which is our primary baseline.
>
> **5. Clarify viewpoint geometry and train/test camera relationship (esp. NTU)**
> - Thank you for pointing this out. We have added a clearer description of the NTU camera setup in Section 4.1.
>
> **6. ETRI: explain “cross-subject only” and consider adding a cross-view protocol**
> - The ETRI "cross-subject" benchmark is similar to others where the data is split by subjects (including all the viewpoints for each subject) into train and test
> - Adding a cross-view evaluation benchmark on ETRI would be valuable, however due to lack of public implementation of our primary baselines as well as lack of pre-training resources, we leave this for future work.
>
> **7. Deepen the analysis of masking ratio vs number of views (“information leakage”)**
> - The optimal masking ratio for MV2MAE is lower (0.7) than that in standard VideoMAE-style settings (0.9) because of multiple reasons. (i) More information is needed per view to infer geometry, (ii) The datasets we use contain videos with motion in a small region with static background and stationary cameras, as opposed to Kinetics/SSv2 datasets which contain moving cameras and significant motion in foreground and background. (iii) MV2MAE uses a lower capacity backbone model ViT-S as opposed to ViT-B
> - The ablation for masking ratio is provided in Table 7.
>
> ### Minor / Other
> - Thank you for the feedback - we have made the updates in the revised paper.
> - The comparison with ViewCLR would require their public implementation which is not available.
> - We have added a *Broader Impact* section in the revised manuscript. Thank you for the suggestion.
> - We have added a mention of Seo et. al. in the *Related Work* for broadening the relevance outside action recognition.

---

> > ### Comment · Reviewer_PaoQ · 2026-01-05
> > **Comment on authors rebuttal**
> >
> > Thanks for the  rebuttal and for making several clarifications in the revision. In particular, adding the explicit test-time protocol (5 temporal clips × 10 crops) to the main tables and clarifying the NTU camera setup improves readability.
> >
> > However, several of my origina;l concerns remain:
> >
> > 1) You note that Table 9 (cross-view decoder off) corresponds to a single-view baseline. This is helpful, but it is currently buried in the ablation section. Please show this baseline in the main results tables (NTU-60/120/ETRI), alongside MV2MAE (and ideally VideoMAE), and make the compute matching explicit (e.g., same #optimizer steps / total tokens / total frames processed). As written, it’s still hard to rule out “extra multi-view data/compute” as a contributor when reading the main comparisons.
> >
> > 2) The 5x10 multi-crop protocol is a very heavy inference-time budget. Even if ViewCLR cannot be re-run without code, please report MV2MAE results under a single-clip, single-crop setting (and ideally VideoMAE too, since its implementation is available) to contextualize the practical inference cost.
> >
> > 3) Stating that parameter counts for other methods are “not available” is not sufficient. Many of the compared methods/backbones do have reported architectures or can be estimated from the paper/implementation. At minimum, please add a small table (main or appendix) listing backbone + parameter count (and ideally FLOPs) for the key baselines in your main comparison tables, along with resolution/#frames and the test-time crop/clip budget. This would strengthen the SOTA/fairness claims.
> >
> > 4) The paper claims MV2MAE is faster/more memory efficient than ViewCLR, but still provides no numbers. If ViewCLR cannot be run, then either (a) soften/remove the claim, or (b) provide a clearly stated quantitative comparison/estimate (peak GPU memory, it/s, GPU-days), plus your measured numbers for MV2MAE. Also, since ViewCLR is presented as the primary baseline, the paper would benefit from a clearer discussion of why MV2MAE succeeds where ViewCLR underperforms (especially in transfer).
> >
> > 5) Only reporting ETRI cross-subject is not very informative for a paper focused on cross-view learning, especially since ETRI has 8 synchronized viewpoints. Even if full baseline replication is hard, It would have been good to add a simple ETRI cross-view protocol (train on a subset of cameras, test on held-out cameras) and at least compare MV2MAE vs your single-view baseline / VideoMAE. If you truly cannot add this, please explicitly acknowledge the limitation and soften cross-view-related claims.
> >
> > 6) Table 7 (mask ratio) and Table 10 (#source views) are useful individually, but do not address the result I asked for.
> >
> > 7) The blurry/rasterized figures still appear in the compiled PDF; please re-export as pdf and add to the tex file.

---

> > > ### Author Response · Authors · 2026-01-16
> > > **Response to Reviewer PaoQ**
> > >
> > > We thank the reviewer for detailed follow-up, and recognizing the updates in the revision.
> > >
> > > - Thank you for this suggestion. The experiment in Table 9 is presented in the Ablation and Analysis section as we are ablating the cross-view decoder, while keeping the motion-weighted loss as in our final approach. Due to compute constraints, this ablation is performed on a single benchmark. Also importantly, the VideoMAE results reported in the main tables (Table 1, 2, 3) correspond to the single-view and compute-matched setting, without using the motion-weighted loss. We match the number of optimization steps for these comparisons. We have updated the paper to mention this.
> > >
> > > - The baselines listed in the paper all report results in the multi-crop setting to ensure temporal and spatial coverage. We add the single-clip, single-crop results to the paper in Ablation and Analysis section, and observe that for NTU dataset where actions are spatially localized, multi-crop testing is essential.
> > >
> > > - We have added a table comparing the parameter counts for key baselines in the appendix.
> > >
> > > - We have removed the claim of faster training compared to ViewCLR in the revised submission.
> > > - We thank the reviewer for this suggestion and agree that an explicit ETRI cross-view evaluation would further strengthen the analysis, given the availability of 8 synchronized viewpoints. Performing careful pre-training and fine-tuning experiments for a newly defined setting would require substantial compute which is beyond our current compute budget. Accordingly, we explicitly acknowledge the absence of an ETRI cross-view evaluation and soften cross-view-related claims for ETRI, clarifying that such conclusions are primarily supported by NTU-60, NTU-120, NUCLA, and RoCoG benchmarks.
> > > - Performing experiments on the full grid of hyperparameters would require pre-training individual models, requiring significant compute budget which is currently not available to us.
> > > - We have updated Figure 3, 4. Thank you for the note.

---

### Decision · Action_Editor_rpPj · 2026-02-27

**Recommendation:** Accept with minor revision

**Additional Comments:**

The paper received mixed reviews. Two of the reviewers are positive and in support of acceptance.

Reviewer sUVt (Leaning Accept):
"... findings justify the claims being made"
"... datasets studied in this paper are typically approached using the skeleton modality, so the use of RGB to model motion is a noteworthy deviation from standard practice"

Reviewer eiaw (Accept):
"The authors have provided a concise and straightforward rebuttal addressing these points (adding required references, adding results in Tables for consistency, clarification on experiments)."

However, Reviewer PaoQ (Leaning Reject) expressed some concerns about the details of the compute-matched "single-view" baseline experiments.

"... the compute-matched “single-view” baseline and key contextual details (backbone/params, inference budget, and what exactly is compute-matched) are not presented in a way that makes the core cross-view benefit immediately unambiguous from the main results"

"... the absence of an explicit ETRI cross-view protocol (despite the dataset having 8 viewpoints) limits the breadth of the paper’s central cross-view generalization story"

The AE thus recommends Accept with minor revision. In the revision, the authors should fully address the remaining concerns from Reviewer PaoQ. This includes
- the details of the compute-matched single-view baselines
- an explicit ETRI cross-view protocol
As suggested, this paper will benefit from more transparently contextualized comparisons.

**Audience:**

Yes

**Audience Explanation:**

Yes, the AE believes that the community will be interested in the motion-aware multi-view MAE.

**Claims And Evidence:**

Yes

**Claims Explanation:**

The paper proposed a self-supervised learning method from synchronized multi-view videos based on the masked autoencoder framework. The two claimed contributions are
- 1) cross-view reconstruction task that leverages a cross-attention-based decoder and
- 2) a controllable motion-weighted reconstruction loss

The paper validates the learned representations on multiple datasets and reports promising performance gains over other self-supervised methods.

---

> ### Author Response · Authors · 2026-03-30
> **Thank you for the comments**
>
> Dear Action Editor,
>
> Thank you for recommendation of *Accept with Minor Revision* and for the detailed summary of the reviewers’ feedback. We appreciate the time and effort from you and the reviewers, and have revised the manuscript accordingly to address reviewers concerns.
>
> Regarding the reviewer’s suggestion to include an explicit ETRI cross-view evaluation protocol, we would like to note an important limitation after contacting the dataset authors: the ETRI dataset used in our original experiments is no longer publicly available for download or access. As a result, we are unable to extend experiments on ETRI to define new cross-view splits.
> To address this limitation responsibly in the revision, we:
> - Explicitly acknowledge that we only evaluate on cross-subject benchmark in the paper,
> - Clarify that our cross-view generalization conclusions are primarily supported by NTU-60, NTU-120, NUCLA, and RoCoG experiments,
> - Soften claims related specifically to ETRI cross-view evaluation,
> - Ensure that all other reviewer requests (compute-matched single-view baseline clarification, parameter/backbone comparison table, single-clip single-crop evaluation, etc.) are fully addressed.
>
> Please let us know if this approach is acceptable for the minor revision.
> Thank you again for your guidance.

---

> > ### Comment · Action_Editor_rpPj · 2026-04-01
> > **Minor revision**
> >
> > Thanks for the note. Yes, it is okay.